# Alzheimer’s Disease in Illinois: Analyzing Disparities and Projected Trends

**DOI:** 10.3390/geriatrics10050132

**Published:** 2025-10-14

**Authors:** Temitope Adeleke, Aston Knelsen-Dobson, Sean McGinity, Kyle M. Fontaine, Benedict C. Albensi, Banibrata Roy, Aida Adlimoghaddam

**Affiliations:** 1Department of Neurology, Dale and Deborah Smith Center for Alzheimer’s Research and Treatment, Neuroscience Institute, Southern Illinois University School of Medicine, Springfield, IL 62702, USA; tadeleke77@siumed.edu (T.A.); kfontaine95@siumed.edu (K.M.F.); 2Department of Psychology, University of Manitoba, Winnipeg, MB R3T2N2, Canada; knelsena@myumanitoba.ca; 3Center for Clinical Research, Southern Illinois University School of Medicine, Springfield, IL 62702, USA; smcginity43@siumed.edu; 4Department of Pharmaceutical Sciences, Barry & Judy Silverman College of Pharmacy, Nova Southeastern University, Fort Lauderdale, FL 33328, USA; balbensi@nova.edu; 5Division of Neurodegenerative Disorders, St. Boniface Hospital Albrechtsen Research Centre, Winnipeg, MB R2H 2A6, Canada; 6Department of Pharmacology & Therapeutics, Max Rady College of Medicine, University of Manitoba, Winnipeg, MB R3E 0T6, Canada; 7Dwyer School of Nursing, Widener University School of Nursing, One University Place, Chester, PA 19013, USA; broy@widener.edu; 8Department of Medical Microbiology, Immunology and Cell Biology, Southern Illinois University School of Medicine, Springfield, IL 62702, USA; 9Department of Pharmacology, Southern Illinois University School of Medicine, Springfield, IL 62702, USA

**Keywords:** dementia, Alzheimer’s disease, gender, age, Illinois, statistics

## Abstract

Alzheimer’s disease (AD) is a growing public health issue disproportionately affecting adults 65 years and older. This growing trend is accompanied by rising economic, social, emotional, and physical costs, both for patients and their caregivers. As the U.S. population ages, understanding disparities in AD prevalence particularly by gender and age has become increasingly important, particularly in high-burden states like Illinois. This review focuses on gender and age disparities in AD, with a specific emphasis on Illinois. This review integrates national and global trends with state-specific projections and explores modifiable and non-modifiable risk factors that may contribute to these disparities. We analyzed projections from the Illinois Department of Public Health and the Alzheimer’s Association to assess AD prevalence by gender and age across Illinois’ 102 counties from 2020 to 2030, disaggregated by gender and age. Rates were compared with U.S. and global trends. Risk factors such as diabetes, education, access to care, and socioeconomic status were reviewed in the context of these disparities. Women consistently show higher AD prevalence across age groups and regions, with the greatest increase in cases is projected among adults aged 75 to 84 years, particularly in regions with higher women populations and social vulnerability. If unaddressed, risk factors like lower education, rural residency, and limited healthcare access may worsen these disparities. Addressing them requires focused public health efforts that combine early screening, caregiver support, and regional resource allocation. Illinois serves as a case study for targeted interventions applicable to broader national strategies.

## 1. Introduction

Alzheimer’s disease (AD) is the most common form of dementia that significantly impacts millions of individuals worldwide and poses a growing threat to aging populations. The survival rate of dementia varies on several factors, including type of dementia [1,2], and it is characterized by the progressive deterioration of cognitive functions, such as memory, reasoning, and communication [3]. Despite advancements in medical research, the exact cause of dementia remains elusive. However, two key non-modifiable risk factors, age and gender, consistently emerge as significant contributors to AD prevalence [3]. Globally, women represent nearly two-thirds of AD cases, and prevalence increases sharply with age, particularly after 75 years [4]. As the U.S. population continues to age, and women outlive men on average, the burden of AD is expected to rise disproportionately among these groups. Yet, these disparities are rarely examined in the context of regional projections. Illinois ranks among the top 10 states for AD prevalence in adults aged 65 and older, offering a unique case study to examine the intersection of age, gender, and Alzheimer’s burden at the state level [5]. Although risk factor distributions are often summarized at the national level, exploring their state-level variation provides new insights into how local demographic and environmental conditions may amplify Alzheimer’s risk in specific regions. To better understand and address these disparities, this review focuses on the types of disparities (age, gender, race, socioeconomic status, and geography), how they are measured (e.g., prevalence per 100,000, diagnosis rates), the accuracy and limitations of those measurements and the mechanisms by which these disparities are generated. The mechanisms include structural inequities, biological factors like APOE4 genotypes, and social determinants such as education, access to care and marital status. This review integrates national and global prevalence data with Illinois-specific projections from 2020 to 2030. Additionally, we evaluate modifiable and non-modifiable risk factors, including diabetes, race, environmental exposures, and healthcare access, and discuss how their regional prevalence and impact may differ from national trends. By doing so, we aim to better understand the multifaceted nature of AD disparities and highlight the need for targeted healthcare policies and interventions that reflect local demographics and needs [6]. With this in mind, AD management can become more personalized, centered on early screening, risk factor modification, and support for both patients and caregivers, further reducing the burden and progression of this disease [7]. This paper is a narrative review that integrates peer-reviewed literature and publicly available datasets to examine gender and age disparities in AD, with a focus on Illinois. Sources were selected based on relevance to national, global, and state-level trends between 2000 and 2024.

## 2. Methods

This narrative review synthesizes relevant publications and public health data from the Alzheimer’s Association, the CDC, the WHO, the Illinois Department of Public Health (IDPH), and peer-reviewed journals. Literature was included based on its relevance to AD burden, risk factors, and disparities, particularly those affecting age, gender, race, and geography in Illinois. This is not a systematic review and does not follow PRISMA criteria. Key metrics and data sources. Prevalence rates were calculated as: Prevalence = (Total number of people diagnosed with AD in Illinois)/(Total Population in Illinois) × 100,000. The standardized prevalence rates were adapted from the 2020 Alzheimer’s Disease Facts and Figures report published in *Alzheimer’s & Dementia*, the journal of the Alzheimer’s Association [8]. The population data were sourced from the 2010 U.S. Census and the Chicago Health and Aging Project (CHAP), a population-based study of chronic health conditions of older people [9]. The source of the cohort-component model used for population projections and demographic breakdowns by age and sex for all 102 Illinois counties came from the Illinois Department of Public Health (IDPH) Population Projections Report of 2023 Edition [10], released on May 2024, covering Illinois, Chicago and Illinois counties by Age and gender from 1 July 2020 to 1 July 2035. This IDPH projection was developed using U.S. Census Bureau data. Trend analysis. Trends in AD cases by sex and age were projected for 2020, 2025, and 2030 using a demographic cohort-component projection model: P_1_ = P_0_ + B − D + NM, where P_1_ is the projected population at the end of the period, P_0_ is the base population at the beginning of the period, B is the resident births during the period, D is the resident deaths during the period, and NM is net migration (In-migration–Out-migration) during the period. The cohort component model, adapted for Illinois, roughly works as follows: for every five-year projection period, the base population, disaggregated by five-year age groups and gender, is “survived” to the next five-year period by applying the appropriate survival rates for each age and gender group; next, net migrants by age and sex are added to the survived population. The population under 5 years of age is generated by applying age-specific birth rates to the surviving women in childbearing age (15 to 49 years). Estimated AD cases, adapted to Illinois and derived using standardized age-based prevalence rates: 3% (age 65–74), 17% (75–84), and 32% (85+), applied to each county’s demographic data. Data visualization and reporting. Figure 1 represents regional definitions of the Illinois Department of Human Services (IDHS). Additionally, we created line graphs showing regional trends by gender (Figure 2) as well as heat maps to depict geographic AD prevalence (Figure 3 and Figure 4), and comparative tables summarizing national, state, and regional AD prevalence and projections (Table 1, Table 2, Table 3 and Table 4).

## 3. Global Trends in Age and Gender Disparities

AD, the sixth leading cause of death overall and the fifth among those aged 65 and older, is recognized by the World Health Organization as a global public health priority [4,16]. It is the most common form of dementia globally, with rising prevalence largely driven by aging populations. In 2023, the World Health Organization estimated that more than 55 million people are living with dementia globally and that number continues to rise by over 10 million new cases each year [11,17]. Without interventions, the number of people with AD and related dementia could grow significantly, with estimates reaching potentially 106 million by 2050 [18]. The burden of AD is highest in high-income countries, yet the most rapid increase is expected in low-income regions where life expectancy is expected to rise. Despite misconceptions that dementia is more prevalent in developed countries, the Delphi consensus study found that while prevalence rates are higher in high-income nations, the number of individuals living with dementia is 3 to 4 times greater in developing countries [11]. Reasons for such reported low prevalence in developing regions can be attributed to a variety of factors, including underdiagnoses of mild dementia, lower life expectancy, reduced environmental risk exposure, lower levels of underlying comorbid diseases, and social or cultural stigmas that limit reporting [11,19]. One study revealed that the Tsimane and Moseten people of the Bolivian Amazon had the lowest prevalence of dementia worldwide. With a prevalence of almost 1%, researchers believed this may have been due to their engagement in high physical activity, such as fishing, hunting, and gathering, their diet, and their low prevalence of cardiovascular disease, obesity, and diabetes [20]. In contrast, other indigenous people of Australia and Guam have a higher prevalence of dementia, and it is believed this is due to not being as isolated from industrialized countries and having adopted the lifestyle of their non-indigenous neighbors [20]. In 2015, China was reported to have the highest population of people affected by dementia at 6 million people, followed by Western Europe and then North America [11]. According to the Alzheimer’s Society of Canada, over 747,000 Canadians are currently living with dementia [21]. This number is projected to exceed 1.7 million by 2050 [22]. One systematic analysis of AD and other dementias from 1990 to 2016 showed that both Nigeria and Ghana had the lowest age-standardized prevalence of dementia [23]. Though the age-specific prevalence of dementia is lower in less developed regions such as Africa and Latin American countries, the number of affected people in these regions is expected to increase as well by 235–393% [11]. The Alzheimer’s Disease International reports that in Africa, the annual number of new dementia cases has increased by over 300,000. Additionally, their findings indicate that the number of individuals living with dementia in Africa is expected to grow from more than 2 million in 2015 to over 7.5 million by 2050. Currently, Latin American countries have about half the number of people living with dementia compared to North America, but this gap is expected to close by 2040 [11]. One meta-analysis found that among all the Latin American countries, Cuba, Puerto Rico, Colombia, and Mexico were found to have the highest frequencies of dominantly inherited AD [24,25]. When compared to the United States and Western Europe, it is thought to be highly influenced by the founder effect, early colonization periods creating reduced genetic variation and a history of inbreeding within large extended families [25]. Additionally, an epidemiological report indicated that in 2016, there were over 700,000 individuals aged 65 or older with dementia in South Korea, with this number expected to grow as the population continues to increase [26]. This number is anticipated to increase significantly due to the aging population and rising life expectancies. Currently, an estimated 8.8 million Indians older than 60 years have dementia, and by 2050, the number of individuals with dementia in India could exceed 14 million [27,28]. Rapid urbanization and increasing levels of air pollution in cities like Delhi have raised public health concerns, with growing evidence linking poor air quality to cognitive decline and dementia risk [29]. Similar trends are emerging in Latin American cities such as São Paulo, Brazil, where urbanization and environmental pollution may also be contributing to the rising dementia burden [30]. As the global population continues to grow, dementia cases are projected to increase rapidly. It is essential to factor these projections into strategies for managing rising dementia rates and shaping future healthcare policies, particularly in developing countries where the surge is expected to be significant. Preparing medically, economically, and emotionally will be crucial to effectively support and manage the growing number of dementia cases worldwide. These global patterns are not just useful for international comparison, but they offer a valuable lens through which to understand local disparities. The same themes, such as limited healthcare access, uneven educational attainment, social vulnerability, and environmental exposures that drive dementia risk worldwide are also evident in Illinois. Primary care providers, including geriatricians, are in high demand rural Illinois counties such as Johnson and Cass, where PCP-to-patient ratios are 13,460:1 and 12,770:1 versus the state average of 1260:1. Combined with shortages of neurologist and limited geographic access, these gaps create major barriers to timely diagnosis and care for older adults [31,32]. Studies have linked lower cognitive performance and higher dementia rates to southern U.S. states with lower educational attainment; similar patterns can be seen in some rural and underserved parts of Illinois, where educational resources and healthcare infrastructure are limited [33,34]. In rural Illinois counties such as Brown and Massac, where high school graduation rates are just 68% compared to the state average of 87%, and where poverty and limited healthcare access compound these disparities, the same link between low education, socioeconomic disadvantage and dementia risk observed in Latin America and Africa is evident [35,36]. Similarly, urban pollution in cities like Delhi and São Pauloparallel risks in Cook County, Illinois, which ranks among the top five most polluted U.S. counties and consistently exceeds the U.S. Environmental Protection Agency’s thresholds for fine particulate matter (PM2.5) [37]. These comparisons help illustrate how global dementia risk factors manifest in localized forms across Illinois, reinforcing the importance of applying global health insights to state-level intervention strategies that target the most vulnerable populations.

## 4. National Trends in Age and Gender Disparities

Similar trends in dementia projections are also evident in the United States. In January 2024, the U.S. Census Bureau projected the population to reach 335,893,238, with the number of Americans aged 65 and older expected to grow from 56 million in 2020 to 88 million by 2050 [8]. This increase in the aging population is anticipated to lead to a higher incidence of dementia. AD affects nearly 10% of Americans aged 65 and older, with an estimated 6.7 million people currently living with the condition [8]. By 2050, this number is projected to reach 12.7 million (Table 1) [3]. The prevalence of AD in the United States varies according to demographic factors [5]. A study examining AD across all 50 states found that individuals aged 85 and older had the highest risk compared to those aged 65 to 69. Additionally, they found women are at a 1.13 times higher risk of developing AD than men, and African American and Hispanic individuals also experience increased risk levels [5]. In the United States, states with larger populations of elderly individuals, especially those aged 85 and older, women, and minorities, such as Maryland and New York, exhibit higher prevalence rates of Alzheimer’s. For example, Maryland’s high Alzheimer’s prevalence of 12.9% is linked to its significant elderly and African American populations [5]. Additionally, counties like Miami-Dade, Baltimore, and the Bronx report some of the highest prevalence rates due to their demographic profiles of African Americans and Hispanic Americans. In 2022, the Centers for Disease Control and Prevention (CDC), a key U.S. public health agency, reported that Mississippi had the highest rate of mortality due to AD, with a rate of less than 50%. Meanwhile, California had the highest number of deaths from AD [5,38].

In 2023, a meta-analysis designed to figure out which states were at a higher risk for dementia based on the area in which individuals live, their birthplace, and associated sociodemographic factors was conducted [39]. Results from every area show dementia is significantly associated with the southern regions of the state [39]. Lower cognition overall is more prominent in southern states where education is poorer than in other areas [40]. Furthermore, a 2024 study on a subset of the data from the Health and Retirement study showed once again that those living in the southern United States are at greater risk for dementia, as education levels and cognitive function in later life are often lower [41]. Lower cognition, overall, is more prominent in southern states where education is poorer than in other areas [40]. Not only does region affect the risk of dementia, but research has also shown a correlation with rural settings. Rural residents were found to be at higher risk for AD and related neurodegenerative disorder mortality than individuals in larger metropolitan areas [42,43,44]. These national trends highlight how broad sociodemographic and geographic factors, such as aging, race, education, and environment, shape AD risk across populations. These same patterns of disparity are reflected within Illinois, particularly in rural counties such as Alexander, Franklin, Gallatin, Hamilton, Hardin, Jackson, Johnson, Massac, Perry, Pope, Pulaski, Randolph, Saline, Union, White, and Williamson—collectively known as the Lower Mississippi Delta Region of Illinois—have fewer healthcare facilities per capita compared to urban regions [45,46]. This disparity contributes to limited access to specialized care and treatment options. Similarly, urban counties like Cook County show higher prevalence among African American and Hispanic populations, paralleling national findings in cities like Baltimore, MD and Miami-Dade, Florida. Likewise, the national trend showing increased Alzheimer’s mortality in states with large African American populations parallels patterns observed in urban regions like Chicago and surrounding counties [47].

Drawing these connections between national and Illinois-specific data helps contextualize local disparities and supports the development of more precise, evidence-based interventions tailored to regional needs. As one of the top states in AD prevalence among older adults, Illinois reflects the broader challenges identified at the national level. We see this particularly in rural, minority, and aging communities. In addition to these demographic and geographic trends, understanding how incidence, prevalence, and mortality interact helps clarify where interventions may be most urgently needed [5]. A rising incidence in older adults contributes to greater disease prevalence, while elevated mortality may signal delays in diagnosis or gaps in care quality. These metrics together provide a more complete picture of disease burden and can guide targeted health planning at both state and national levels.

## 5. Illinois-Specific Trends in Age and Gender Disparities

Illinois mirrors national AD trends but shows clear regional variation. According to the Alzheimer’s Association, the projected number of Americans living in Illinois who suffer from AD will increase by 13% between 2020 and 2025 [8]. In a study analyzing the prevalence of AD in the United States, Illinois was among the top 10 states with the highest prevalence of AD in the United States [5]. Illinois was also found to be among the top 5 states in the United States with the highest prevalence of AD in people aged 65 years and older [5]. Table 1 provides an overview of dementia statistics for the state of Illinois compared to the rest of the country as well as globally.

As these numbers continue to climb, the toll of an aging population will continue to impact potential caregivers and those responsible for their healthcare. Illinois residents aged 65 or older are projected to grow from 2.1 million in 2020 to 2.7 million by 2030, with Chicago’s older population expected to increase from over 320,000 in 2020 to more than 540,000 by 2030 [48]. To adequately care for this population, Illinois would need to increase its number of geriatricians from 200 in 2021 to over 1500 geriatricians by 2050 and expand its home care aide workforce to over 118,000 by 2030 [4].

Cook County in Chicago, Illinois, reports the highest number of Alzheimer’s cases in Illinois [5]. In 2020. It reported an estimated prevalence of 765,189 cases compared to 159,658 in DuPage County. However, when adjusting for population, per capita prevalence reveals that rural counties may bear a similar or even greater burden due to higher proportions of elderly residents and limited healthcare infrastructure. Of all 3142 counties in the U.S., Cook County ranked second highest in AD prevalence [5].

To better understand the geographic distribution and prevalence of AD across the state, the Illinois Department of Public Health (IDPH) divided the state into five regions (Figure 1). Standardizing AD prevalence per 100,000 people allows for a clearer comparison of disease intensity between counties of different sizes. While Cook County leads in raw case counts, rate-adjusted data reveal emerging hotspots in smaller, aging, or under-resourced areas that might be otherwise overlooked. Figure 2 illustrates that the predicted regional trends of AD cases are higher in women across all five IDPH Regions from 2020 to 2030, suggesting that women will be disproportionately affected in the coming years. Figure 2 highlights regions 3 and 4, which cover central Illinois and several smaller counties, as having the highest projected prevalence rates, particularly among aging women populations. Though Figure 4 shows that some counties in regions 2 and 3 exhibit particularly high rates, the broader burden in regions 3 and 4 still drives up regional averages. The data suggest that regions 2 and 3 may experience the largest increase in AD cases over the next decade. However, these projections are based on raw prevalence estimates without statistical testing; therefore, the observed differences should be interpreted cautiously. As shown in Figure 2, the prevalence rates in regions 3 and 4 are similar to or slightly higher than the statewide average. This can primarily be attributed to the elevated diagnosis rates among women in these regions, as indicated in Figure 3. In Figure 4, although region 4 does not appear to have as many high-prevalence counties as region 3, this may be due to an uneven distribution of AD across counties in region 4. Instead, the impact is concentrated in specific counties within that region, which may require more county-specific interventions. While Regions 3 and 4 may benefit from AD care programs in general, certain counties within these regions are bearing a particularly heavy burden. Thus, community-specific targeted interventions such as community outreach programs, specialized healthcare facilities, and Alzheimer’s community support groups may be beneficial for these heavily impacted counties. As prevalence rates are expected to rise significantly by 2030, regions 3 and 4 may experience greater demands on both caregivers and healthcare systems. Addressing these regional disparities early on could help alleviate future strain on these communities. Although Region 5 has an overall lower prevalence of AD compared to regions 3 and 4, Figure 2 and Figure 3 show that the gender disparity is still present, with women being more affected. Interestingly, Figure 3 reveals a mixed pattern across region 6. While most counties in region 5 show a decrease in the gender disparity over the years, a few counties, such as Randolph, Gallatin, Fayette, and Jefferson, stand out as emerging hotspots. These counties exhibit a darker shading over time, indicating a worsening disparity among women. While region 5 may not be experiencing a high prevalence of cases, these specific counties could be driving localized increases, possibly due to region-specific shifts in demographic risk factors. Unlike the other regions where the prevalence is uniformly increasing, Region 5′s trends highlight the importance of implementing county-level interventions to better allocate resources and address local disparities.

These regional and county-specific findings, summarized by Figure 2 and Figure 4, indicate that the issue of AD is widespread across high-density areas rather than confined solely to the largest population centers, a trend consistent with other major urban areas in the United States. This pattern is likely influenced by factors such as healthcare access, infrastructure, and urban living conditions. The elevated prevalence of AD in these areas can be attributed to various demographic factors, such as a higher percentage of at-risk adults, along with environmental factors. Dementia rates are often higher in metropolitan areas, where access to quality healthcare services and higher levels of education and awareness contribute to the diagnoses and management of Alzheimer’s [49]. In these regions, better access to healthcare can lead to more diagnoses being reported, whereas areas with limited access may experience delayed diagnoses. Additionally, the incidence of dementia is often higher in urban areas, where levels of air pollution and chemical toxins are elevated, and this exposure has significantly risen over the last decade [50]. Chicago and its metropolitan area rank third for population in America. Illinois consistently ranks among the top five most polluted states nationally, with Chicago reporting PM2.5 levels significantly higher than the U.S. average. This is concerning, given studies linking long-term air pollution exposure to increased Alzheimer’s risk through inflammatory and vascular pathways [51]. According to the 2022 Illinois Air Quality Report, Chicago has the highest PM2.5 pollution levels in the state of Illinois [37]. Andrews et al. (2024, 2025) concluded that neural atrophy is linked to long-term exposure to air pollution. Their findings highlight significant associations between prolonged exposure to fine PM2.5 and increased propensity to develop anxiety and depression [52,53]. It is proposed that air pollution affects the central nervous system because of its effects on increased proinflammatory mediators and reactive oxygen radicals [54]. Air pollution caused by exhaust gas was also found to accelerate the accumulation of Aβ-42, a known cause of neuronal dysfunction. Therefore, long-term exposure to air pollution can lead to Aβ-42 peptide and neurofibrillary tangle formation in the brain [50]. Additional factors such as socioeconomic inequality, education levels, healthcare access, and lifestyle factors such as high stress, poor diet, and lack of physical activity interact with environmental exposures to drive disparities [55]. Cook County is ethnically and socioeconomically diverse, with research indicating that certain ethnic groups, such as African Americans and Hispanics, may have a higher risk of AD [47]. Together, these patterns highlight that AD burden in Illinois is not limited to high-density urban centers, but spans both rural and suburban areas in complex ways. Understanding these regional and county-specific trends is essential for developing targeted public health policies and allocating resources equitably.

## 6. Risk Factors

While the Alzheimer’s Disease Facts and Figures report provides a foundational overview of national-level risk factors, this review focuses on how those same risk factors manifest uniquely across Illinois. By comparing Illinois-specific prevalence, socioeconomic conditions, and environmental exposures to national averages, we aim to highlight state-level nuances that contribute to regional disparities in AD burden. This state-based lens may help inform more tailored public health responses in Illinois and other states with similar demographic or geographic risk profiles.

## 7. Non-Modifiable Risk Factors Contributing to Disparities

### 7.1. Age

Age is a primary risk factor for AD and other dementias. As the population of the United States continues to age, the number of individuals at risk for developing dementia is expected to rise correspondingly [4]. According to the United States Census Bureau (2022), the population of individuals in Chicago aged 65 or older was 13.3% [56]. Table 1 shows that AD cases in Americans aged 65+ are expected to rise from 5.8 million in 2020 to 12.7 million by 2050, with global dementia cases expected to reach 152.8 million. The number of Illinois residents living with and affected by dementia is dramatically increasing, especially for those aged 65 and above. The largest increase is expected to occur in the 75–84 age group with a projected rise of 52% by the year 2030. Nationally, the U.S. also anticipates a similar rise in cases, with an estimated increase of 2.47 million from 2020 to 2030. This dramatic increase is largely driven by the aging 75–84 and 85+ populations, with a projected increase of 58.7% in the 75–84 alone, as reflected in Table 2 This data highlights a parallel rise in AD both statewide and nationally, showing a strong correlation between advancing age and the increased risk of the disease. It underscores the growing impact of AD as the population ages, emphasizing the urgent need for resources to support aging populations facing this disease. As the global population grows, and the 65+ population grows, so does the risk of developing dementia. These projections emphasize the critical need for effective interventions and support systems to address the escalating risk of dementia as the population ages, focusing on early screening initiatives and expanding long-term care services. Such interventions could possibly help mitigate this burden and the rising numbers in the older age groups.

### 7.2. Gender

Gender is another significant risk factor associated with dementia, with numerous studies indicating that women are more likely to develop the condition when compared to men [57]. According to the Alzheimer’s Association’s 2023 report, among the 6.7 million Americans aged 65+ living with AD, roughly 4.2 million are women, and 2.5 million are men [4], meaning approximately 63% of those with AD are women. The prevalence of AD among women is expected to rise further. By 2060, projections suggest that 8.22 million U.S. women will have the disease compared to 5.64 million men (Table 3). This disparity is likely due to a combination of genetic, social, and biological factors [58,59,60]. This disparity may be attributed to a combination of genetic, social, and biological factors. One key genetic factor is the APOE ε4 allele, the strongest known genetic risk factor for late-onset AD. While APOE ε4 increases AD risk in both genders, studies have shown that women who carry the allele may experience a greater risk and earlier onset compared to men [58]. This heightened vulnerability may be due to hormonal interactions or gender-specific brain metabolism differences. Social factors, such as women’s increased caregiving roles, can add stress and contribute to cognitive decline, further raising their risk [60]. Biologically, research suggests that hormonal differences, particularly the decline in estrogen during menopause, may impact brain aging and memory, compounding the risk of developing AD in women [59]. Illinois mirrors national patterns, with women disproportionately affected by AD. Figure 2 depicts the predicted trends of AD cases by gender across Illinois, showing a higher number of cases in the state and in each region at every time point. As the number of AD cases increases over time in all regions, the rates are higher for women in each region, reflecting a disparity. Figure 3 also illustrates the distribution of AD cases in each county in the state of Illinois by gender, revealing that most counties have a higher proportion of AD cases in women. Table 2 further underscores this disparity, with women showing a higher prevalence rate than men in every region. These findings align with global data indicating that women account for two-thirds of dementia cases [4]. While women have a higher overall prevalence of AD, men also experience significant risk, particularly those who are single or widowed [60]. It has been shown that men who lack the emotional and social support of a spouse or partner are more vulnerable to cognitive decline [59]. Among divorced individuals, divorced men experience greater risks for dementia when compared to divorced women [60]. Though divorce may reduce social networks in general, men often have smaller social networks and rely more heavily on their spouses as their confidants [60]. Thus, the loss of a spouse can lead to greater social isolation and increased dementia risks compared to women who usually have a broad network of friends and relatives as their confidants. In Illinois, t men with AD cases are also rising, suggesting that social vulnerability may be a critical yet underrecognized risk factor for men. The demographic of men in Illinois is rapidly growing, and this increase in AD cases among men suggests that social factors are an underappreciated contributor to this disease’s progression. However, there are some gender-specific drawbacks to traditional marriages specifically for women. Women are more psychologically and physiologically vulnerable to marital stress than men [60]. This is because women tend to take on more responsibilities for maintaining social connections, offering emotional support to their spouses, and managing their spouse’s health behaviors. All factors that may increase the risk of dementia among married women [60]. When compared to married respondents, the unmarried groups tended to have lower education levels, lower income, less likelihood of exercise, and higher proportions of current smokers [61]. Those who are unmarried were also found to be at a higher risk of underdiagnosis in clinical setting due to the absence of collateral information from a spouse or partner [61]. These findings highlight the importance of addressing gender-specific risk factors and social determinants of health in relation to dementia. Tailored interventions are needed such as bereavement support and social engagement programs for men, and strategies that address biological risk and caregiving burden for women.

### 7.3. Race and Ethnicity

Lower rates of cognition are correlated to education, socioeconomic factors, and race, which ultimately put some individuals at a higher risk of developing dementia. Lower cognitive function has been shown in racialized groups [62,63,64]. People with dementia are often racialized as African American or Hispanic, which is influenced by a combination of socioeconomic, cultural, and health factors [3]. Socioeconomic disparities such as lower income levels, reduced access to quality healthcare and education, and systemic racism all affect cognitive health outcomes [3]. While these factors may not directly cause cognitive impairment, they significantly influence it. Studies on dementia healthcare disparities have identified differences in services and treatment, misdiagnosis, and medication use among the racial minority population [65]. For instance, African American and Hispanic individuals living with dementia are more likely to live in under-resourced nursing homes with limited advanced care planning compared to non-Hispanic White individuals [65]. Additionally, post-diagnosis, African Americans and Hispanics are more likely to discontinue anti-dementia medications and experience higher hospital mortality rates, which may explain the trend towards more aggressive behaviors and costly end-of-life care [65]. Stereotypic threats also impact the diagnosis of dementia among African American individuals by affecting their performance on cognitive tests. The research found that when African American undergraduates at Stanford University were told that a test measured intellectual ability, they performed worse compared to white students with similar SAT scores [47]. In contrast, when the same test was described as a simple laboratory problem-solving task, the performance between groups was equal [47]. This suggests that when African American patients take cognitive assessments like the MoCA, knowing it evaluates their memory or thinking may trigger stereotype threat and lead to underperformance, potentially resulting in false positives or misdiagnosis of dementia. Social and cultural differences can also influence the perceptions and responses to cognitive decline among different ethnic groups [19,47]. Some groups considered dementia-related changes to be a normal part of aging, which can delay the recognition and diagnosis of dementia when symptoms become severe. In other cultures, like those in Taiwan and parts of Latin America, cognitive decline is viewed as disgraceful and is not made public, making it difficult to seek management and support [19,47]. It is often difficult to accept when the ethnic elder is seen as the provider of the family in all aspects. For example, one study showed that African American elders who were evaluated at dementia centers often reported a shorter duration of illness at the time of initial diagnosis, yet showed more advanced cognitive dysfunction and impairment at the time of initial diagnosis [47]. This could have been due to reporting bias, denial, or lack of understanding of the disease, causing them to delay disclosure to providers. These disparities are especially relevant in Illinois, where cities like Chicago have large African American and Hispanic populations that are disproportionately affected by such social and healthcare inequalities. According to the United States Census Bureau (2022), in 2022, the White population of Chicago was 42.4%, the African American population was 28.8%, the Hispanic population was 29%, and the American Indian/Alaska Native population of Chicago was 0.7% [56]. Given these demographics, the increased prevalence of dementia among African American and Hispanic populations suggests a disproportionate impact on Chicago communities, especially in neighborhoods with high concentrations of these groups. A review of epidemiological studies found that African Americans and Hispanics have higher rates of dementia, while Native Americans have lower rates than Whites [47]. The higher prevalence and incidence in African American, Hispanic, and Native American populations translates to a higher chance that their families from these populations will serve as caregivers for a relative with dementia, which can lead to poorer health outcomes for the caregiver [65]. In Illinois, the burden on caregivers may be especially pronounced in urban areas like Chicago, where access to culturally competent support services remains uneven.

African Americans have significantly lower vitamin D levels due to the melanin in the skin causing insufficient vitamin D production, especially when living at higher northern latitudes where the UVB radiation dose is lower [66]. Vitamin D deficiency is associated with an increased risk of AD and dementia, with studies showing that serum levels below 10 ng/mL correspond to a 31% higher risk of dementia and a 33% higher risk of AD [66]. With a 15-to-20-fold higher prevalence of severe vitamin D deficiency in African Americans, this could contribute to their increased risk of developing dementia [66]. This risk is relevant in northern cities like Chicago, where limited sunlight exposure further worsens vitamin D deficiency in African Americans. However, there is controversy regarding whether vitamin D supplementation helps prevent or slow cognitive decline. Some animal studies suggest that Vitamin D supplementation can improve cognitive outcomes if initiated early before amyloid plaques have formed, while others indicate that prolonged supplementation may worsen AD progression by affecting the vitamin D receptor pathways [67]. Additionally, one study found no benefits for vitamin supplements above 50 nmol/L in adults [68]. Research also proposes that vitamin D deficiency may be more of a consequence rather than a cause of AD, as lower vitamin D levels have also been observed in other aging-related diseases [67]. Despite these findings, further larger clinical trials generalizable to diverse populations may be needed to determine optimal levels of vitamin d supplementation and the timing for intervention.

Racial minorities are also underrepresented in clinical trials for AD, partly due to the mistrust of the federal government and the institutions conducting the research. Indigenous populations in Canada and the United States are less likely to participate in randomized control trials (RCTs) for AD, cultural language differences, feeling excluded, false promises, lack of engagement, knowledge, and community outreach as potential reasons [16,69]. In 2021, the Alzheimer’s Association found Native Americans to be at a higher risk for AD and other dementias compared to White or Asian Americans, facing barriers like limited healthcare access, delayed diagnoses, and caregiver stress [70]. Up to 1 in 3 Native American Elders may develop dementia, with cases among those aged 65 and older expected to increase fourfold by 2060 [69]. Although 92% value culturally competent care, only 49% have it, and 61% find affordability a barrier. Concerns about research bias are widespread, with 40% believing it influences outcomes and only 65% confident that a cure for AD would be publicly shared. Additionally, Indigenous populations face significant barriers to participating in research, including RCTs, leading to less accurate data on AD prevalence [69]. More research is still needed for minority groups to accumulate more data regarding race/ethnicity and the risk of AD or other dementias. It is crucial for future dementia research to address Illinois’s racial and ethnic diversity, especially in urban areas like Chicago, where unique needs and disparities impact these communities.

### 7.4. Modifiable Risk Factors Contributing to Disparities

#### 7.4.1. Diabetes

Diabetes is continually identified as a leading cause of death in America and within Illinois. In addition, it is a recognized risk factor for dementia [71]. According to IDPH, 84 million Americans have prediabetes, while in Illinois, approximately 1.34 million adults have diabetes and about 3.6 million have prediabetes [35]. This represents a slightly higher rate than the national average, especially in central and southern regions where diabetes prevalence overlaps with underserved populations, potentially compounding AD risk in these areas. National studies show diabetes increases AD risk by up to 50%, but this effect may be amplified in Illinois due to demographic and healthcare disparities [72]. AD is referred to as “type 3 diabetes” to reflect the association between the two diseases [72,73]. Both type 2 diabetes and AD share common genetic and environmental risk factors as well as underlying pathology [72]. Several hypotheses have been proposed to explain this association. One study suggests that elevated plasma glucose and insulin are associated with reductions in plasma amyloid precursor protein, with insulin specifically playing a role in decreasing Aβ neurotoxicity [74]. Another suggests that the insulin resistance underlying the pathophysiology of diabetes has led to peripheral metabolic disturbances and vascular brain injury, which are contributing to neuronal damage [72]. Additionally, individuals with diabetes have a higher risk of stroke and small-vessel diseases, which can increase the likelihood of developing dementia [47]. Insulin plays a crucial role in neuroprotection, and its absence has been associated with neurodegeneration and impaired hippocampal plasticity [75]. Insulin resistance may also disrupt the blood–brain barrier’s permeability, therefore interfering with synaptic plasticity and cognition [75]. Furthermore, insulin receptors have been found to be concentrated in brain regions involved in cognition, including the hippocampus, amygdala, parahippocampal gyrus, thalamus, and caudate putamen, highlighting its importance in maintaining brain health [76,77]. Epidemiologic studies have shown that type 2 diabetics are at a 50–100% increased risk of developing diabetes [78]. The risk for diabetes in relation to stroke-associated dementias is 2 times greater in African Americans and Hispanics than in Whites [71]. Among African Americans and Hispanics, 33% of the risk associated with stroke-associated dementia was attributable to diabetes, compared with 17% among whites [47]. Native Americans in the United States are nearly three times as likely as non-Hispanic White adults to be diagnosed with diabetes, which increases their risk for dementia compared to other subpopulations [69]. According to the Alzheimer’s Association, 37% of Medicare beneficiaries with AD who are 65+ also have diabetes [4]. The cost of comorbid AD and diabetes is higher than for diabetic patients without AD [4]. In 2022, patients who had comorbid AD and diabetes averaged just under $28,000 per year in Medicare payments, while diabetic patients without AD spent an annual average of less than $18,000 [4]. Therefore, more research is required to determine the linkage between AD and diabetes.

#### 7.4.2. Access to Healthcare

Limited access to healthcare services significantly impacts dementia diagnosis and management, affecting both the timing and the effectiveness of disease management. Compared to national averages, Illinois shows higher regional variation in dementia care access, particularly in southern and western counties with few neurologists For geriatric care centers [31]. This geographic disparity may contribute to delayed diagnosis and worse outcomes among rural populations. Illinois is experiencing a healthcare provider deficit, and by 2030, projections suggest a state-wide shortage of 6200 doctors. Rural counties are most affected by this deficit, with 81 of Illinois’ 102 counties experiencing a county-wide shortage of primary care professionals, while ten counties near Chicago and Springfield have been able to maintain an adequate supply. Restrictive occupational licensing laws have exacerbated the shortages and limited economic opportunities and access to care. Approximately one quarter of Illinois’ workforce requires an occupational license. Licensure and legal regulations have cost Illinois nearly $15.1 billion in misallocated resources and the loss of an estimated 135,000 jobs. In 2025, a new law was introduced to ease barriers to licensure for foreign-educated doctors, which is expected to grow the healthcare workforce and improve access to care throughout Illinois.

Factors contributing to restricted access include limited income, shortage of clinicians skilled in dementia management, inadequate resource allocation, lack of health policies, and residential segregation [79]. These barriers often result in delays in diagnosis, exacerbating dementia progression. One study highlighted that restricted healthcare access leads to increased emergency department visits and poor continuity of care, further worsening dementia outcomes [80]. Similarly, another study found that individuals in underserved populations face challenges managing dementia alongside comorbid conditions like diabetes and hypertension, due to inadequate healthcare access [79,81]. Better geospatial healthcare access has been found to lead to better health outcomes [32]. The lack of regular accessible care often causes diagnoses to be made at a later stage, reducing opportunities for early intervention. Competing health system priorities also contribute to these issues [19]. For instance, there is a focus on the health of younger people, pregnant women, and infectious diseases. These competing priorities often overshadow the needs of the aging population [19]. This can affect resource allocation decisions. Additionally, underserved areas often lack neurologists and neurophysiologists, and have the insufficiency of diagnostic tools such as neuroimaging, cognitive assessment instruments, and biomarker testing that further hinders standardized dementia care [19]. A survey noted that some clinicians view working in dementia as less appealing, exacerbating the shortage of specialists [3]. Moreover, policy goals and inadequate healthcare financing often result in plans that exist on paper but lack the resources or stakeholder support for implementation [19]. Addressing these disparities by improving healthcare access is crucial for ensuring timely diagnoses and effective management of dementia and its comorbidities. Early intervention and diagnosis of AD can ultimately improve healthcare quality and quality of life for dementia patients and their families.

#### 7.4.3. Education

Educational attainment is another risk factor influencing the rate of dementia, with lower levels of education correlating with higher risks [34]. This correlation reflects disparities in literacy rates and overall quality of education. According to the US Census Bureau (2022), 86.6% of Chicagoans aged 25 or older have a high school diploma or higher, and only 42.4% hold a Bachelor’s degree or above [56]. While the national average for bachelor’s degree attainment is around 37%, certain Illinois counties fall significantly below this benchmark, particularly in rural and minority communities. This disparity may heighten vulnerability to cognitive decline given the strong link between lower education and reduced cognitive reserve [82,83]. Research indicates that reading comprehension and literacy have a stronger relationship with cognitive test performance than years of education alone, suggesting that quality of education plays an important role in cognitive health [34]. The 2024 Lancet Commission report also highlights the importance of being cognitively, physically and socially active in midlife and later in life as even with little education, cognitive activity makes a difference [84]. Cognitive reserve is defined as “the adaptability (i.e., efficiency, capacity, flexibility) of cognitive processes that helps to explain differential susceptibility of cognitive abilities or day-to-day function to brain aging, pathology or insult” [85,86]. Education is thought to contribute to cognitive reserve that can help delay the onset of neurogenerative conditions like dementia. Unfortunately, individuals with less enriched educational experiences often have limited cognitive reserves, which can impact the brain’s ability to delay the onset of dementia symptoms [47]. Additionally, unfamiliarity with medical concepts can lead to misdiagnosis or misunderstanding of cognitive decline, highlighting the need for better education on these topics. In many native American communities, cognitive decline might be viewed through a cultural or spiritual perspective rather than as a medical condition like AD, potentially leading to delayed recognition and treatment [69]. Dementia can also be misconstrued as a normal part of aging but with more education, this lack of awareness can be rectified. A systematic review found that lower education attainment levels increased the risk of developing dementia in older adults, with 58% of reviewed studies supporting this link [34]. Moreover, education levels are closely tied to income and socioeconomic status with research showing that higher socioeconomic status is associated with longer periods without dementia [87]. Studies have shown that African Americans often score lower on neuropsychological tests compared to their white counterparts even after adjusting for education levels [47]. Despite these findings, further research is still needed to fully understand the impact of education on dementia development.

#### 7.4.4. Lifestyle Factors

The 2020 Lancet Commission report identified 12 modifiable risk factors for dementia across various stages in life. With early intervention of addressing these risk factors, there is potential to prevent or reduce about 40% of dementia cases. [55,84]. In early life, lower education levels contribute to an increased risk while in midlife, factors such as hearing loss, hypertension, traumatic brain injury, excessive alcohol consumption and obesity play a significant role [55]. Later in life, tobacco use, depression social isolation, physical inactivity, diabetes, and air pollution tend to contribute to dementia risk [55]. Additionally, lifestyle factors such as diet, exercise, and sleep are significant contributors to the development and progression of dementia [3,55,88]. One study showed that when Japan transitioned from its traditional diet to a Western diet, dementia rates increased by 6% [88]. Research has consistently shown that foods high in saturated fats such as meats, eggs, and high-fat dairy products tend to increase the risk of dementia [88]. This is often attributed to the way saturated fats can contribute to inflammation, oxidative stress, and vascular damage, all of which are factors in the development of neurodegenerative diseases [88]. However, adherence to a diet rich in grains, fruits, vegetables, and fish has been shown to reduce the risk of dementia [89,90]. A clinical trial in Spain found that a Mediterranean diet supplemented with either extra virgin olive oil or nuts improved cognitive performance, whereas those on the control diet experienced cognitive decline [58]. These benefits of the Mediterranean diet were independent of gender, age, energy intake, and cognition-related variables, including educational level, APOE ε4 genotype, and vascular risk factors. The study highlighted that the inclusion of olive oil and nuts, rich in antioxidants and anti-inflammatory agents, likely contributed to these cognitive benefits by counteracting oxidative processes in the brain, improving cerebrovascular blood flow, and enhancing neurogenesis [58]. Many studies have shown that the Mediterranean-DASH Intervention for Neurodegenerative Delay (MIND) is a diet that has been shown to have protective effects on cognitive decline [91]. The MIND diet is a combination of the Mediterranean diet and the DASH diet with an emphasis on the consumption of green leafy vegetables, nuts, berries, olive oil, and less intake of foods high in saturated fat and sugar [92,93]. One study found that high consumption of berries, specifically blueberries and strawberries, have been associated with a slower cognitive decline by almost 2.5 years [94]. These fruits are known to be high in antioxidants such as anthocyanidins and flavonoids, which can cross the blood–brain barrier and localize to the hippocampus for neuroprotection [94,95]. Strawberries are the most abundant source of pelargonidin, an anthocyanin, which has an inverse relationship with phosphorylated tau tangles among those without the APOE ε4 allele but not in APOE ε4 carriers [95]. A study funded by the National Institute on Aging examined the association between the MIND diet and AD pathology in post-mortem brain tissues [89]. This study found that both the MIND and Mediterranean diets were associated with less AD brain pathology and higher cognitive function, especially those who incorporated more green leafy vegetables into their diet [89,90]. There is a strong association with regular consumption of 1 daily serving of green leafy vegetables such as spinach, kale, collard greens, and lettuce, with slower cognitive decline [96]. These vegetables are rich with nutrients such as lutein, folate, vitamin K, and the flavonoid kaempferol [96]. Lutein has been found to attenuate neuroinflammation and oxidative stress, while folate has been found to inhibit tau phosphorylation and APP, PS1, and Aβ protein levels [96].

Aside from proper diet, higher fitness levels are considered neuroprotective and can significantly mitigate age-related cognitive decline [97]. One clinical study found that individuals aged 40–59 with an overweight BMI of 25–30 had a 35% higher chance of developing AD later in life compared to those with a normal BMI of 20–25. In contrast, those in the same age group with a lower BMI of less than 20 had only a 2% increase chance of developing the disease later in life [98]. Typically, the hippocampus shrinks 1–2% yearly in adults without dementia [99,100]. However, in a randomized controlled study, moderate-intensity aerobic exercise demonstrated a substantial impact on hippocampal volume in older adults [97]. Over a one-year period, the aerobic exercise group exhibited an increase in the volume of the left and right hippocampus while the stretching control group experienced a decline in hippocampal volume over the same interval [97]. Incorporating a balanced diet and regular exercise can significantly reduce the risk of dementia, with evidence showing that both positively impact cognitive health and neuroprotection.

Sleep is another modifiable lifestyle factor that can impact the risk of developing dementia [3]. About 70% of patients with early-stage dementia experience sleep disturbance, which includes issues with sleep duration and quality [101]. One proposed mechanism is that sleep deprivation causes excessive neuronal excitability, leading to greater amounts of amyloid-beta (Aβ) accumulation, which contributes to the pathogenesis of AD [3,101]. Another theory is that the glymphatic system, which helps clear toxic aggregates like Aβ, becomes less effective with age [101]. During sleep, the activity of the glymphatic system increases, likely due to an increase in the interstitial space as astroglia cells shrink [101]. Therefore, sleep disruptions can impair this system and potentially promote the development of AD. Among those with AD, disrupted sleep can lead to agitation, aggression, and poor daily functioning [3]. Shorter sleep durations have also been associated with loss of brain volume through hippocampal degeneration due to decreased synaptic plasticity, decreased neurogenesis, and changes in neuronal excitability [101]. One study found that sleep durations less than 5 h and greater than 9 h were found to be associated with twice the risk of AD [101]. A cohort study including about 1.5 million subjects found that those with any sleep disorder had a 17% higher risk of dementia compared to those with no sleep disorders and the risk was highest during the first 5 years of their sleep disorder diagnosis [102]. With this finding, it is thought that sleep disorders can not only be seen as a risk factor but also an early symptom of dementia [102,103]. Additionally, obstructive sleep apnea (OSA) has also been linked to AD pathophysiology [3] as acute hypoxia has been shown to cause an increase in tau phosphorylation and chronic hypoxia has been shown to increase Aβ plaques [101]. OSA has been associated with a 2 to 6 times increase in the risk of mild cognitive impairment or dementia [101]. The treatment of sleep disorders could be one of the preventative measures to improve dementia symptoms [104]. Treatment with continuous positive airway pressure (CPAP) has been seen to improve the cognitive function of people with mild to moderate AD [105]. Reducing the risk of dementia not only increases the number of healthy years in life but also shortens the duration of the symptoms for those diagnosed with dementia [84]. Evidence shows that interventions during midlife are important, but some risk factors originate across different stages in life or are associated with socioeconomic status [84]. While early intervention is ideal, addressing these risk factors at any stage of life is beneficial as long as it is maintained [84]. With proper modification of these factors, such as diet, exercise, and sleep, there can be protective effects against cognitive impairment.

#### 7.4.5. Marital Status

Marital status is considered a proposed modifiable risk factor for dementia, which can be influenced by social, psychological, and economic resources [60,61]. According to the marital resource model, marriage is associated with better health outcomes and longer life expectancy, whereas being unmarried can have detrimental effects on a range of health outcomes [60]. Married individuals benefit from unique resources that those in platonic relationships or cohabiting do not experience. These unique advantages include greater economic resources, healthier behaviors, and enhanced social interactions [60].

Economic resources contribute to the 13%-18% increased risk of dementia among divorced, widowed, and never-married individuals but not among cohabitors [60]. These resources include wealth pooling and specialization which contribute to better health outcomes by providing improved nutrition and access to medical care [60]. Unmarried individuals lack these economic advantages which can negatively impact their cognitive health. In 2013, more than 3.3 million adults aged 50 years and older were cohabiting in the United States [60]. Though cohabitors can benefit somewhat from shared living expenses, most do not wealth pool or specialize to the same extent as married couples. Cohabitation seems to be a long-term alternative to marriage among older adults, but studies have shown that cohabitors have a higher dementia risk than married individuals, likely due to lower commitment levels and lack of legal protection [60].

The stress model further reveals the negative impacts of marital disruption on cognitive health [60]. Divorce and widowhood are significant stressors that can lead to financial and emotional distress, directly affecting cognitive function [60]. The stress associated with marital disruption can lead to unhealthy behaviors such as smoking, heavy drinking, and reduced physical and social activities, along with chronic conditions like cardiovascular diseases and diabetes, all linked to cognitive impairment and dementia [60,61]. Another study found widowed individuals to have a 20% higher risk of developing dementia, while lifelong single individuals were found to have a 42% higher risk compared to married individuals [61]. The increased risk amongst widowed individuals may be partly explained by the stress-induced cognitive decline following the loss of a spouse, while the risk for lifelong single individuals may be due to different social engagement and physical health behaviors [61]. However, with societal changes, the normalization of remaining single, and the decreased likelihood of experiencing the stress of marital disruption, this risk amongst lifelong singles may decrease over time [61]. Potentially mitigating some of the risks associated with other unmarried groups. Contrary to such findings, divorced individuals were found to have the highest risk of dementia, not widowed individuals [60].

The sociopsychological resources associated with marriage also play a crucial role in cognitive health. Married individuals are seen to enjoy increased social engagement, support, and integration, which are linked to better health outcomes and reduced dementia risk [60]. Social engagement and larger social networks help improve cognitive reserves and regular interaction with a spouse provides daily cognitive stimulation and enhances neural plasticity, maintaining cognitive reserves [60]. Cohabiters, however, receive fewer sociopsychological benefits due to lower commitment levels and a lack of institutional legitimacy [60]. This can result in less social support from friends and family compared to married individuals, which may contribute to a higher risk of cognitive decline.

With the rise in the number of unmarried individuals in the United States and complex marital histories, understanding the role marital status plays in dementia is important. Marriage has been shown to play a protective role against the development of dementia while divorce and widowhood have been noted to increase the risk. Integrating individuals’ marital status into dementia risk assessments and interventions can serve as a vital piece in providing comprehensive dementia care.

### 7.5. Economic and Caregiver Burden

AD imposes a dual burden on families, healthcare systems, and society through both caregiving demands and economic strain. These burdens are closely intertwined as emotional and physical stressors for caregivers often translate into financial sacrifices, while the rising costs of care stretch both private households and public health budgets.

Dementia caregivers and healthcare workers report growing difficulties in managing dementia care in current U.S. healthcare system [3]. The COVID-19 pandemic and widespread healthcare workforce shortages have only exacerbated this crisis [106,107]. As of now, more than 80% of caregiving for individuals with AD or other dementias are provided by informal caregivers [4,108]. Informal caregiving is usually provided by unpaid family members or friends who often step into this role unexpectedly, without formal training or support [3]. The most common reasons family and friends take on the burden of serving as a caregiver include physical proximity to the patient, a sense of obligation or love and wanting their loved ones in a comfortable environment such as their home [109]. These caregivers frequently bear a variety of responsibilities, including medication management, navigating healthcare systems, coordinating appointments, addressing behavioral symptoms, and providing emotional support [4]. With such a long duration of AD, the intensity and duration of these caregiving tasks often require caregivers to reduce their work hours or take time off, resulting in serious financial and psychological consequences [11]. Caregivers commonly experience stress related to managing care costs, securing resources, and advocating for patients [3,109]. Many report experiencing chronic anxiety, depression, substance abuse, and even physical illness such as stroke, hypertension, and diabetes as a result of prolonged caregiving stress [4]. Nationally, more than 11 million Americans serve as unpaid dementia caregivers, including over 300,000 in Illinois [3,4]. Between 2015 and 2021, rates of depression amongst caregivers in Illinois rose by 29%, yet only about half of the caregivers reported discussing these challenges with a healthcare professional [3]. Despite this, caregivers acknowledge that resources like a 24/7 helpline, care coordination services, and education on AD progression would significantly help [3]. However, most primary care providers report not feeling equipped to guide families or manage the complexities of dementia care [4].

The financial toll of caregiving can be understood through two main categories: formal (direct) costs and informal (indirect) costs [110,111]. Formal costs refer to out-of-pocket expenses and services for which money is exchanged, such as physician visits, hospitalizations, long-term care facilities, and paid home health services [110]. In contrast, “informal care costs are defined as the cost of the time spent by family or friends caring for a person without payment, and they can include the costs of stopping or reducing work” [112]. Informal costs also encompass the emotional and physical strain experienced by caregivers [109].

In 2019, the average cost of caregiving for someone with AD or related dementias exceeded $42,000 per year [113]. If all individuals received the full extent of needed care, total costs would have surpassed $230 billion in 2019 and are projected to reach $404 billion by 2050 [113]. When broken down further, the average annual caregiving cost for a person with dementia is over $14,000 compared to a little over $10,000 for individuals without dementia [114]. Informal caregiving, in particular, carries major hidden costs, including lost wages, career disruptions, and increased healthcare utilization by caregivers themselves [4]. The financial impact of AD begins well before a formal diagnosis is made. A 2016 study comparing individuals with AD, mild cognitive impairment (MCI), and matched controls found that healthcare expenses were already higher for those with AD before diagnosis [115]. The most significant costs occurred during the first year after diagnosis, averaging over $27,000, compared to $20,000 for those with MCI [115]. These findings highlight how progressive cognitive decline can generate substantial healthcare expenditures even in the early stages. It is important to note that older individuals with MCI are at risk of progressing to AD or related dementias, and the cost for these individuals tends to rise sharply as their symptoms worsen [116,117].

Globally, dementia-related care accounts for $818 billion USD annually. In the U.S., costs reached $345 billion in 2023 and are projected to reach $1.6 trillion by 2050 [4]. This leaves dementia patients and their families with an estimated debt of about $390,000 [4]. Although Medicare and Medicaid cover a portion of these expenses, about 36% of these costs are paid out-of-pocket by caregivers. These include the cost of food, medications, and other informal caregiving responsibilities [4]. In 2019, Medicare expenses for dementia patients were 3 times higher than those without dementia, while Medicaid costs were 23 times higher [109]. The economic strain is compounded by the combined weight of formal and informal costs, with the latter often becoming more burdensome as the disease progresses [109]. Informal caregiving costs increase significantly during the mild and moderate stages of AD, eventually surpassing formal care costs as caregiving responsibilities intensify. One study found that while informal care costs in the early stage were initially lower, they eventually overtook formal care costs as caregiving responsibilities intensified [118]. The mild stage of AD is often the most expensive phase, both financially and emotionally, as families assume greater care duties [118].

In Illinois, this trend is reflected in recent projections. A 2023 study in Illinois reported that Medicaid payments for AD patients aged 65 and older are expected to rise by more than 20% between 2020 and 2025, increasing from $1.7 million to $2.2 million [4]. These increases are in line with national trends and are driven by the rising prevalence of AD and an aging population. Additionally, dementia care informal costs surpass those associated with other major chronic conditions, such as cancer and cardiovascular disease, averaging $83,000 annually, compared to $38,272 for other illnesses [109]. This discrepancy highlights the uniquely prolonged, intensive, and costly nature of AD care.

Taken together, these findings and discrepancies highlight the intense need for systemic reforms and proactive support structures. Delaying the onset of AD by even 1 to 5 years could reduce healthcare payments by up to 39%, suggesting that investment in prevention, caregiver support, and early intervention is not only humane but economically necessary [109]. Without such action, the rising financial and emotional toll of AD will become unsustainable for individuals, families, and healthcare systems alike.

### 7.6. Interventions and Recommendations

Throughout the various studies of AD and dementia research, there have been many suggestions for interventions for families, patients, and physicians, future research, and initiatives that can improve the quality of life. The National Institute on Aging has a website that allows individuals and physicians to find AD research centers (ADRC) in their geographical areas [119]. Not every state has an ADRC. There are two ADRCs in Illinois, according to the National Institute on Aging. These ADRCs are located at Northwestern University and Rush University Medical Center [119]. The ADRC is beneficial to the advancement of AD research and clinical trials. The Alzheimer’s Association has an online page which allows patients, their families, and physicians to gain knowledge about various clinical trials and match them based on which may be suited for their case [120].

A prominent theme in research recommendations for AD is early education. This education is suggested to be beneficial for the patient, the caregiver/spouse/family members, and the physician/medical experts. Recommendations include increasing knowledge of minority groups and associated risk factors [64,121].

Another theme in AD research is intervention and community engagement. A pilot study for online chair yoga was recently conducted [122]. The study was designed as a tool to assist those with AD and related dementias in underserved areas. The participants of the online chair yoga resulted in significant improvements across various factors associated with dementia, including cognition [122]. Although their results were found to be not significant when compared to those who played an online game, the results indicate more research is needed with regard to technology and interventions for those who suffer from AD and related dementias [122]. The study indicates the need to develop easily accessible interventions for individuals who suffer from AD and related dementias. Future interventions should include beginner-level use of technology for ease for both the memory-impaired patient and the overworked caregivers. The interventions should also be made easily accessible (such as digital and free) so that rural communities and those of minority groups have access to the interventions. Interventions such as online chair yoga provide individuals with cognitive impairment and AD a sense of community and friendship [123]. Socialization and friendship are known factors of positive outcomes for older adults facing cognitive impairment [123,124].

It is important to remember that the burden of AD falls on many. Research shows burnout can be an unfortunate consequence for a caregiver of someone with dementia, which can also negatively impact the patient [125]. Furthermore, caregiver psychological and mental health also impacts patients’ psychological and mental health [126]. More research is needed regarding the impact of caregiver mental health on patients’ psychological health. Other research shows the impact of support groups for caregivers [127]. The review found peer group support was a positive factor for caregivers of those with AD [127]. Various reviews provide insight into how future research should include more education and interventions focused on assisting families and caregivers [128,129]. Research also suggests diagnostic and coding policies need to be reviewed and revised to potentially decrease the financial burden placed on patients and their caregivers [130].

It is also important to remember that caregivers include medical professionals. The Alzheimer’s.gov webpage has various resources to assist medical professionals in caring for individuals with AD [131]. Results from a survey of medical professionals dealing with dementia patients suggest that more education is required [132]. The medical professionals’ answers to the survey suggest they lack knowledge of resources available to patients and families [132]. Future interventions should not neglect or overestimate the knowledge of physicians with regard to patient resources. Future interventions should be designed with continuing education for new and established physicians in mind, such as the curriculum developed by Padala et al. in 2020 [133].

## 8. Conclusions

The number of Illinois residents living and affected by dementia is increasing rapidly, particularly among those aged 75–84, indicating where future healthcare demands will intensify. This finding aligns with national trends, further reinforcing the strong association between aging and AD. However, Illinois-specific disparities also emerge particularly by gender, race, and geography, underscoring the need for a more localized understanding of disease burden. This review highlights that while age and gender are universal non-modifiable risk factors, disparities in education, access to care, and environmental exposures like air pollution intensify AD risk in specific Illinois populations. Compared to national averages, Illinois shows elevated PM2.5 pollution, regional healthcare access gaps, and lower educational attainment in certain counties, which together may exacerbate vulnerability to AD.

The predicted regional trends of AD cases are higher in women across all five Illinois IDPH regions from 2020 to 2030, indicating that women will be more significantly affected by the disease in the coming years. Understanding these trends can help prepare for this shift as it emphasizes the need for more resources and support for the rapidly growing 75–84 age-group population, who will soon account for a significant portion of AD patients. While the 85+ age group does not have the highest number of cases, they still represent a significant number of cases and will continue to place a heavy demand on healthcare services because of their long-term healthcare needs. Illinois mirrors national upward trends in terms of prevalence of disease, but regional variations also exist, with Cook County having the highest prevalence of AD when accounting for both gender and age. Table 4 presents a side-by-side comparison of AD projections by age group in both Illinois and the United States, illustrating an upward trend in cases across all age groups, with total U.S. cases expected to surpass 8.5 million by 2030 and continue rising through 2060. Compared to national averages in Table 4, Illinois AD prevalence goes from slightly lower in 2020 (1.78% vs. 1.83%) to nearly matching it by 2030 (2.4% vs. 2.44%), possibly influenced by healthcare access, demographics, and social determinants of health amongst the different counties. As Illinois’s projected AD prevalence nears national averages by 2030, tailored interventions that address regional disparities will be critical. These should include early screening, culturally competent care, increased provider training, and community-based caregiver support. Further, understanding how national-level risk factors behave in Illinois, especially among women, minorities, and rural residents, can drive more equitable and effective health policy initiatives to effectively manage the growing number of dementia cases and mitigate their impact. Ultimately, preparing Illinois’s healthcare infrastructure to handle the rising tide of AD will require coordinated efforts across policy, education, and clinical practice to reduce disparities and improve outcomes for all affected populations.

## Figures and Tables

**Figure 1 geriatrics-10-00132-f001:**
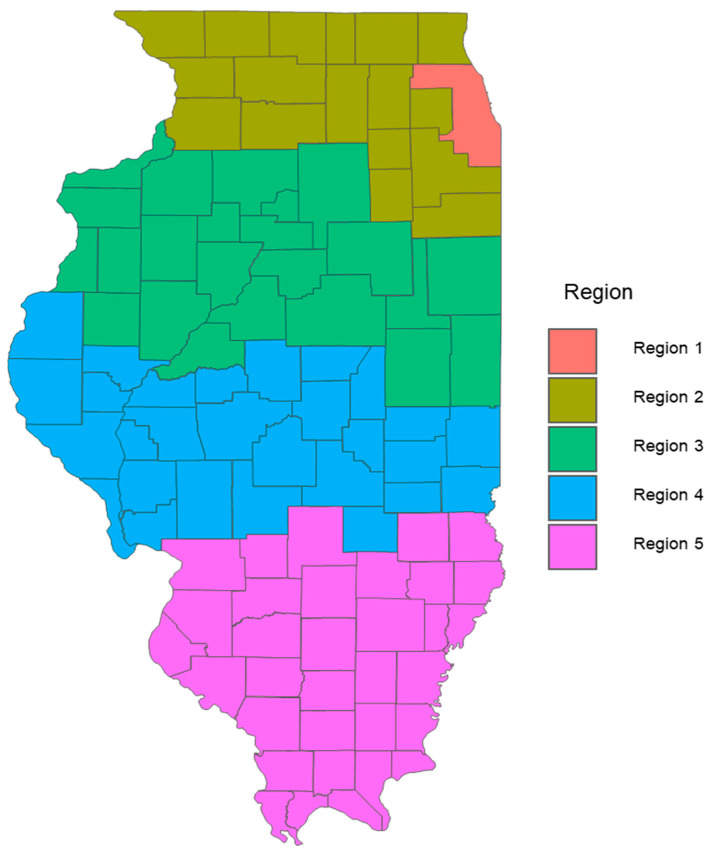
Regional Definitions of the Illinois Department of Human Services (IDHS). This figure illustrates the five regions defined by the Illinois Department of Human Services (IDHS) for analyzing healthcare and demographic data related to Alzheimer’s disease. The regions are delineated to aid in understanding the geographic distribution and prevalence of Alzheimer’s, facilitating targeted public health initiatives and resource allocation. A map is adapted from the IDHS website. https://www.dhs.state.il.us/page.aspx?item=55223 (accessed on 1 January 2020). Region 1: Cook, Region 2: Jo Daviess, Stephenson, Winnebago, Boone, McHenry, Lake, Carroll, Ogle, DeKalb, Kane, DuPage, Whiteside, Lee, Kendall, Grundy, Will, Kankakee, Region 3: Rock Island, Mercer, Henry, Bureau, LaSalle, Henderson, Warren, Knox, Stark, Putnam, Marshall, Livingston, Ford, Iroquois, Vermillion, Champaign, McLean, Woodford, Tazewell, Mason, Peoria, Fulton, McDonough, Region 4: Hancock, Adams, Schuyler, Brown, Cass, Menard, Logan, Dewitt, Piatt, Douglas, Edgar, Clark, Coles, Cumberland, Effingham, Shelby, Moultrie, Macon, Christian, Montgomery, Sangamon, Morgan, Macoupin, Green, Jersey, Calhoun, Scott, Pike, Region 5: Madison, Bond, Fayette, Clay, Jasper, Crawford, Lawrence, Richland, Edwards, Wabash, Wayne, Marion, Clinton, St. Clair, Monroe, Randolph, Washington, Jefferson, Perry, Jackson, Franklin, Hamilton, White, Williamson, Saline, Union, Johnson, Pope, Hardin, Alexander, Pulaski, Massac, Gallatin.

**Figure 2 geriatrics-10-00132-f002:**
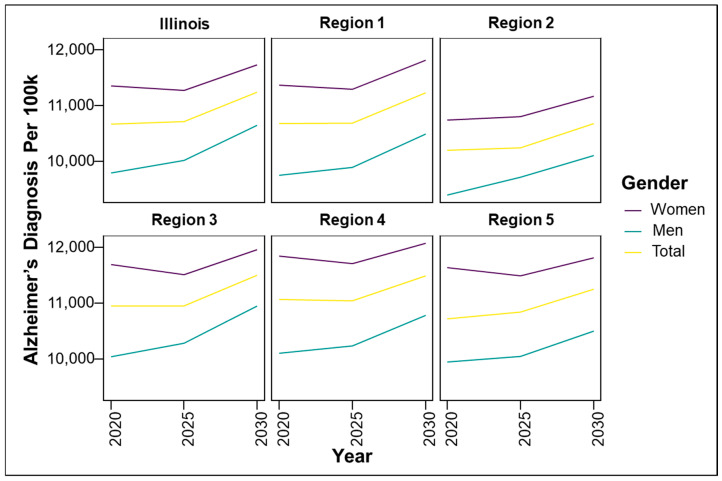
Predicted regional trends of Alzheimer’s disease (AD) cases by gender across Illinois and the five Illinois Department of Human Services (IDPH) Regions from 2020 to 2030.

**Figure 3 geriatrics-10-00132-f003:**
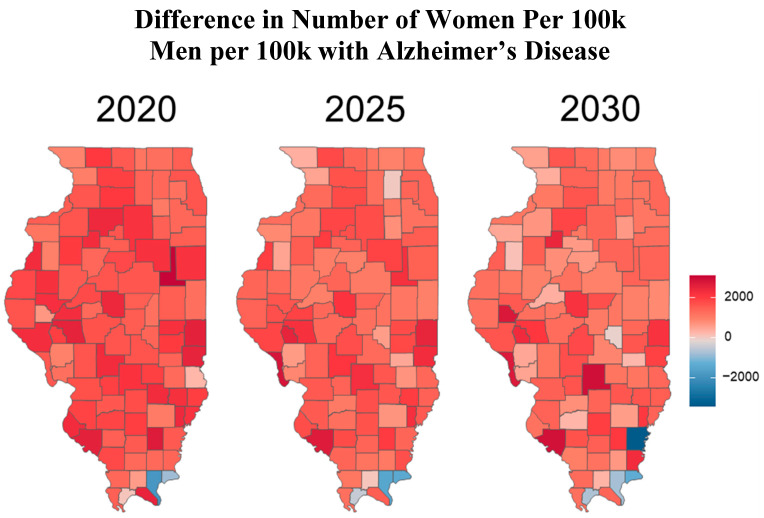
The heat maps above illustrate the difference in Alzheimer’s disease (AD) rates between women and men in each county. This is calculated by subtracting the number of men with Alzheimer’s per 100,000 males from the number of women with Alzheimer’s per 100,000 women. Counties shaded in red indicate a higher proportion of women diagnosed with AD, while those shaded in blue indicate a higher proportion of men diagnosed.

**Figure 4 geriatrics-10-00132-f004:**
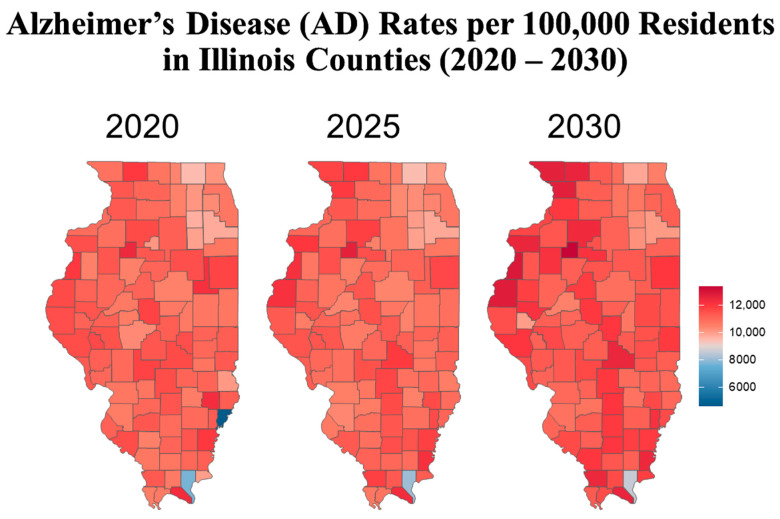
The heat map above shows the prevalence of Alzheimer’s disease (AD) per 100,000 residents aged 65 and older in each county. Areas with the greatest disease burden are represented in deep red, while areas with the lowest burden appear in deep blue.

**Table 1 geriatrics-10-00132-t001:** Dementia statistics overview.

	Previous Count	2023 Count	2050 Count
Global	24 M, as of 2005 [11]	55 million (WHO, 2023) [12]	152.8 M [13]
United States	3.4 M, as of 2002 [14]	6.7 M [4]	12.7 M [4]
Illinois	230,000 as of 2020 [4]	260,000 [4]	

Data are in millions (M).

**Table 2 geriatrics-10-00132-t002:** Number of Alzheimer’s disease (AD) cases per 100,000 individuals by gender across the five Regions of Illinois, as defined by the Illinois Department of Human Services. These Regions are used to assess the geographic distribution and prevalence of Alzheimer’s, supporting targeted public health strategies and resource allocation. For more details about the Region’s specification, refer to Figure 1.

	2020	2025	2030
Illinois			
Women	11,354.20	11,273.86	11,729.42
Men	9791.51	10,016.94	10,646.36
Total	10,664.85	10,710.92	11,241.40
Region 1			
Women	11,366.33	11,291.13	11,819.29
Men	9749.89	9891.67	10,489.27
Total	10,675.04	10,683.38	11,236.67
Region 2			
Women	10,740.23	10,801.77	11,166.73
Men	9395.66	9717.44	10,108.88
Total	10,204.38	10,248.24	10,681.32
Region 3			
Women	11,694.37	11,510.63	11,964.03
Men	10,034.99	10,278.21	10,946.88
Total	10,947.54	10,950.31	11,502.34
Region 4			
Women	11,845.19	11,709.16	12,076.28
Men	10,099.44	10,231.59	10,781.86
Total	11,068.50	11,044.18	11,493.75
Region 5			
Women	11,636.89	11,492.51	11,809.56
Men	9941.05	10,042.98	10,503.48
Total	10,719.21	10,840.38	11,249.38

**Table 3 geriatrics-10-00132-t003:** Alzheimer’s disease projections by gender in the United States [15].

	2024	2030	2060	Total Cases
Women	4.23 M	5.17 M	8.22 M	17.62 M
Men	2.71 M	3.37 M	5.64 M	11.72 M

Data are in millions (M).

**Table 4 geriatrics-10-00132-t004:** Comparison of AD Projections in Illinois and the United States (2020–2030) [15].

Age	Illinois 2020	U.S. 2020	Illinois 2025	U.S. 2025	Illinois 2030	U.S. 2030
65–74	37,314	1.65 M	42,134	1.88 M	43,950	2.05 M
75–84	103,270	2.18 M	129,183	2.79 M	157,439	3.46 M
85+	85,269	2.24 M	90,948	2.49 M	103,908	3.03 M
Total	225,853	6.07 M	262,265	7.16 M	305,297	8.54 M

Data are in millions (M).

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
