# Peer review of "Alzheimer’s Disease in Illinois: Analyzing Disparities and Projected Trends"

_geriatrics, 2025, doi:10.3390/geriatrics10050132_

Round 1
Reviewer 1 Report
Comments and Suggestions for Authors
The manuscript entitled “Alzheimer's Disease in Illinois: Analyzing Disparities and Projected Trends” by Adeleke et al, presents a comprehensive and well-organized analysis of the current and projected burden of Alzheimer’s disease in Illinois, with particular attention to demographic and geographic disparities. The authors integrate state-specific data with national and global trends and highlight risk factors such as age, sex/gender, healthcare access, education, and air pollution. Overall, the manuscript is timely and relevant, but there are areas that need clarification before publication.
- The paper mixes up sex and gender. It refers to gender but then categorizes people as male and female. Please distinguish between sex as biological traits and gender as social and cultural categories.
- The discussion of race and ethnicity risks reinforcing stereotypes. Attributing disparities to vitamin D levels suggests a biological explanation for race, which is a social construct. The explanation for underrepresentation of minority groups in clinical trials places blame on mistrust rather than on systemic exclusion and harm. The framing should be revised with care, using resources such as the Alzheimer’s Association Inclusive Language Guide.
- The methods section needs clarification. The cohort component model and standardized prevalence rates should include the source and note whether they were adapted for Illinois specifically.
- Some gender findings are repeated in multiple sections. These can be streamlined to make the text more concise.
- The national and global comparisons should be linked back to Illinois more directly. For example, discussions of rural isolation or low education could be tied to specific Illinois counties.
- The manuscript would benefit from proofreading.
With these revisions the paper would be much stronger and suitable for publication.
Author Response
Reviewer 1:
The manuscript entitled “Alzheimer's Disease in Illinois: Analyzing Disparities and Projected Trends” by Adeleke et al, presents a comprehensive and well-organized analysis of the current and projected burden of Alzheimer’s disease in Illinois, with particular attention to demographic and geographic disparities. The authors integrate state-specific data with national and global trends and highlight risk factors such as age, sex/gender, healthcare access, education, and air pollution. Overall, the manuscript is timely and relevant, but there are areas that need clarification before publication.
The paper mixes up sex and gender. It refers to gender but then categorizes people as male and female. Please distinguish between sex as biological traits and gender as social and cultural categories.
The comment has been addressed by reviewing the entire text and ensuring consistent and appropriate use of the term gender to reflect social and cultural roles, while avoiding conflation with sex-based (biological) classifications such as male and female.
The discussion of race and ethnicity risks reinforcing stereotypes. Attributing disparities to vitamin D levels suggests a biological explanation for race, which is a social construct. The explanation for underrepresentation of minority groups in clinical trials places blame on mistrust rather than on systemic exclusion and harm. The framing should be revised with care, using resources such as the Alzheimer’s Association Inclusive Language Guide.
The language is consistent with that used in the referenced paper (Reference: 64 - Ames BN, Grant WB, Willett WC. Does the High Prevalence of Vitamin D Deficiency in African Americans Contribute to Health Disparities? Nutrients. 2021;13(2):499) from which the concept of Vitamin D levels was reported. We simply restated what was previously reported by Ames et al. that African Americans have significantly lower vitamin D levels due to the melanin in the skin causing insufficient vitamin D production, especially when living at higher northern latitudes where the UVB radiation dose is lower (64). This paper has been cited more than 150 times since its publication.
The methods section needs clarification. The cohort component model and standardized prevalence rates should include the source and note whether they were adapted for Illinois specifically.
The comment has been highlighted and addressed on pages 4-5.
Some gender findings are repeated in multiple sections. These can be streamlined to make the text more concise.
The introduction will briefly highlight gender as a non-modifiable risk factor, and the dedicated “Gender” section will carry the full analysis with Illinois-specific projections. Redundant gender findings were integrated into the gender section on pages 15-16.
The national and global comparisons should be linked back to Illinois more directly. For example, discussions of rural isolation or low education could be tied to specific Illinois counties.
Global and national comparison sections have been revised to relate more explicitly to Illinois. Illinois counties and regions where rural isolation, educational disparities, and minority population density mirror the international and national risk factors have been identified on pages 7-9.
The manuscript would benefit from proofreading.
The manuscript has been thoroughly proofread to improve clarity, grammar, and consistency throughout.
With these revisions the paper would be much stronger and suitable for publication.
Reviewer 2 Report
Comments and Suggestions for Authors
The manuscript entitled “Alzheimer's Disease in Illinois: Analyzing Disparities and Projected
Trends” by Adeleke et al provides a narrative review of Alzheimer’s disease disparities, using
Illinois as a case study within global and national contexts. The authors draw on data from IDPH,
CDC, WHO, and other sources to highlight trends by age, gender, and geography, and they
incorporate projections to 2030. The authors also discuss non-modifiable (age, gender,
race/ethnicity) and modifiable (diabetes, healthcare access) risk factors. Overall, the review is
well-structured, timely, and relevant for both scientific and policy audiences.
Before this manuscript can be considered for publication, I recommend the following revisions:
1. Gender disparities are emphasized multiple times (abstract, gender section, Illinois
projections). Please streamline into a single, concise summary to avoid redundancy.
2. The standardized prevalence rates and cohort-component model need more explanation.
Specify the source of these rates (e.g., Alzheimer’s Association, CDC) and whether they
were adapted for Illinois.
3. Strengthen the link between global/national examples and Illinois context (e.g., connect
rural isolation or low education in Illinois counties to parallels mentioned globally).
4. Risk Factors:
o Connect race/ethnicity discussion more explicitly to Illinois demographics (e.g.,
Chicago’s African American population).
o In the diabetes section, add Illinois-specific prevalence data (e.g., 3.6 million
cases) to anchor projections.
o Expand the healthcare access section with state examples (e.g., neurologist
shortages in southern Illinois).
5. Proofreading: Correct typographical errors and standardize terminology (“Alzheimer’s
disease” on first mention, “AD” thereafter). Ensure figure and table references are
consistent.
6. Update environmental data (e.g., PM2.5 levels in Chicago) with the most recent
references where available.
The manuscript addresses an important topic and has strong potential. With the above revisions, it would be suitable for publication.
Author Response
Reviewer 2: The manuscript entitled “Alzheimer' s Disease in Illinois: Analyzing Disparities and Projected Trends” by Adeleke et al provides a narrative review of Alzheimer’s disease disparities, using Illinois as a case study within global and national contexts. The authors draw on data from IDPH, CDC, WHO, and other sources to highlight trends by age, gender, and geography, and they incorporate projections to 2030. The authors also discuss non-modifiable (age, gender, race/ethnicity) and modifiable (diabetes, healthcare access) risk factors. Overall, the review is well-structured, timely, and relevant for both scientific and policy audiences.
Before this manuscript can be considered for publication, I recommend the following revisions:
Gender disparities are emphasized multiple times (abstract, gender section, Illinois
projections). Please streamline into a single, concise summary to avoid redundancy.
Repetitive information was removed or combined, and references to figures and tables were integrated for clarity.
The standardized prevalence rates and cohort-component model need more explanation.
Specify the source of these rates (e.g., Alzheimer’s Association, CDC) and whether they were adapted for Illinois.
The comment has been highlighted and addressed on pages 4-5.
Strengthen the link between global/national examples and the Illinois context (e.g., connect rural isolation or low education in Illinois counties to parallels mentioned globally).
Global and national comparison sections have been revised relate more explicitly to Illinois. Illinois counties and regions where rural isolation, educational disparities, and minority population density mirror the international and national risk factors have been identified on pages 7-9.
Risk Factors:
o Connect race/ethnicity discussion more explicitly to Illinois demographics (e.g.,Chicago’s African American population).
We revised the text to explicitly connect race and ethnicity to Illinois demographics, particularly Chicago. We emphasized how dementia-related disparities affect African American and Hispanic communities locally, and incorporated census data and geographic factors (e.g., vitamin D deficiency in northern regions) to strengthen the Illinois context. The comment has been addressed and the revisions are highlighted on pages 15–17, where we explicitly connect race and ethnicity to Illinois demographics, with a focus on Chicago’s African American and Hispanic populations.
o In the diabetes section, add Illinois-specific prevalence data (e.g., 3.6 million cases) to anchor projections.
The comment has been highlighted and addressed on page 17.
o Expand the healthcare access section with state examples (e.g., neurologist shortages in southern Illinois).
The comment has been highlighted and addressed on page 19.
Proofreading: Correct typographical errors and standardize terminology (“Alzheimer’s disease” on first mention, “AD” thereafter). Ensure figure and table references are consistent.
We have carefully proofread the manuscript to correct typographical errors and have standardized terminology throughout. “Alzheimer’s disease” is now spelled out at first mention, with “AD” used in subsequent references. Additionally, all figure and table references have been reviewed and updated to ensure consistency.
Update environmental data (e.g., PM2.5 levels in Chicago) with the most recent references where available.
Updated PM2.5 data for Chicago using the most recent available sources and revised references accordingly (Page 12).
The manuscript addresses an important topic and has strong potential. With the above revisions, it would be suitable for publication.